# Sasang Constitution Type Combined with General Obesity May Act as a Risk Factor for Prediabetes Mellitus

**DOI:** 10.3390/healthcare10112286

**Published:** 2022-11-15

**Authors:** Younghwa Baek, Siwoo Lee, Kyoungsik Jeong, Eunsu Jang

**Affiliations:** 1KM Data Division, Korea Institute of Oriental Medicine, Daejeon 34054, Republic of Korea; 2Division of Diagnosis, College of Korean Medicine, Daejeon University, Daejeon 34520, Republic of Korea

**Keywords:** Sasang constitution, Soyangin, Taeeumin, general obesity, prediabetes mellitus, risk factor

## Abstract

Sasang constitutional medicine is a traditional customized medicine in Korea that classifies people into four types: Taeeumin (TE), Taeyangin (TY), Soeumin (SE), and Soyangin (SY). This study explored whether Sasang constitution (SC) types combined with general obesity could be risk factors for prediabetes mellitus (pre-DM). This study was cross-sectional and was conducted from November 2007 to July 2011 in 23 Korean medical clinics. In total, 2185 eligible subjects participated. A *t* test, one-way ANOVA with Scheffé’s post hoc analysis, the chi-square test and multinomial logistic regression were used. Significance was indicated by *p* < 0.05. The numbers of participants with normal fasting plasma glucose (FPG) levels and pre-DM were 405 (75.3%) and 133 (24.7%) in the SE, 516 (70.3%) and 218 (29.7%) in the SY, and 590 (64.6%) and 323 (35.4%) in the TE (*p* < 0.001) groups, respectively. There was a significant difference in the proportion of each SC type among people with pre-DM and normal FPG levels in the normal BMI group. The odds ratios (ORs) of the TE type were significantly different from those of the SE type in the crude and Model 1. The distribution of the normal FPG and pre-DM individuals between the obese and normal BMI groups only for the SY type was significantly different. The SY type combined with general obesity had a higher OR (1.846, 95% CI 1.286–2.649) than that combined with normal BMI among participants with pre-DM, and this higher OR remained after adjusting for covariates (OR, 1.604, 95% CI, 1.093–2.354). This study revealed that the TE type might be a risk factor for pre-DM in the normal BMI group, and the SY type with general obesity could be a risk factor for pre-DM compared with the SY type with normal BMI. Accordingly, SC and BMI should be considered when managing pre-DM. To clarify the risk of SC and BMI, further study including epigenetic factors is needed.

## 1. Introduction

Prediabetes mellitus (pre-DM) is defined as an intermediate state of hyperglycemia, with glucose levels above the normal state but below the diagnostic level of diabetes [1]. Pre-DM is becoming increasingly important as it is associated with a high risk of developing type 2 diabetes (T2D) and cardiovascular diseases [2,3]. The prevalence of pre-DM has increased [4], and the prevalence has noticeably increased among middle-aged and older individuals. The prevalence of pre-DM among middle-aged adults aged over 40 years of age was 18–20%, which increased by 9–11% between 2009 and 2016 [5]. Another study found that among adults with unhealthy lifestyles and psychological stress, especially males, the prevalence of pre-DM increased by 1.4–1.9 times [6,7].

Diabetes gives rise to microvascular and macrovascular complications and results in a socioeconomic burden. The American Diabetes Association (ADA) estimated that the national cost associated with diabetes in the United States in 2017 was $237 billion [8]. To reduce this burden of disease, it is necessary to prevent diabetes by managing its risk factors. T2D is known to occurs more frequently in the elderly and specific ethnicities, as well as individuals who are obese, less physically active, have a family history of the disease, or have previously been diagnosed with gestational diabetes mellitus [9,10,11].

Among the risk factors, obesity is a medical condition in which excess body fat has accumulated to the extent that it may have an adverse effect on health, leading to reduced life expectancy and/or increased health problems [12]. Obesity plays a pivotal role in prediabetes and diabetes [13].

Sasang constitutional medicine (SCM) is a traditional customized medicine in Korea that classifies people into four types: Taeeumin (TE), Taeyangin (TY), Soeumin (SE), and Soyangin (SY) [14]. Each constitution has different characteristics in terms of body shape, face, voice, and psychological and physiological aspects [15,16]. Therefore, each element is different from one type to another [17]. On the basis of this rationale, each constitution is susceptible to a specific pathology and drug response. Several studies revealed that hypertension, diabetes mellitus (DM), metabolic syndrome and so on were associated with a specific Sasang constitution (SC) [18,19,20,21].

According to SCM theory, kidney hypofunction and spleen hyperfunction are related to food digestion, and the SY type is associated with hyperactive spleen function; this type is highly susceptible to ‘So-gal’ disease. The symptoms of ‘So-gal’ disease are known to be excessive drinking, urination, and thirst, which are similar to symptoms of DM. Accordingly, medications administered for the treatment of diabetes are usually used for the SY type. Lee revealed that the SY and TE types may be risk factors for DM [19]. However, there has been no research on pre-DM and SC, especially regarding related obesity.

We hypothesized that the SY type combined with obesity is associated with high susceptibility to pre-DM on the basis of ‘longevity and life preservation in oriental medicine’. In this study, we presented clinical evidence of whether the SY type stratified by body mass index could be a risk factor for pre-DM among Koreans.

## 2. Materials and Methods

### 2.1. Data Source and Subjects

This study was cross-sectional and was conducted from November 2007 to July 2011 in 23 Korean medical clinics. We extracted data from the Korean Medicine Data Center (KDC). The KDC contains various clinical and blood parameter data collected from a consortium consisting of Korean and Western medical institutions and community-based cohorts in Korea [22].

The subjects included those who voluntarily agreed to participate in this study among adults who met the criteria for taking SC-specific pharmaceuticals. The exclusion criteria included subjects who were physically unable to follow the instructions of the researcher, those who had a deformation in the measurement location, and pregnant women. Subjects were consecutively recruited via posts on both online and offline boards during the study period [22]. A total of 2688 subjects were enrolled in this study, but 91 were excluded because of missing data. The data of 331 DM patients who were taking medicine to treat high blood sugar were excluded according to the ADA criteria [23]. Furthermore, as the composition of the TY type is fairly rare in Korea, 81 individuals with the TY type were excluded from this study. Ultimately, a total of 2185 subjects were analyzed in this study. A detailed flowchart of the study design is shown in Figure 1.

The Korea Institute of Oriental Medicine Institutional Review Board approved this study (KIOM IRB; number I-0910/02-001), and written informed consent was obtained from each subject.

### 2.2. Diagnostic Criteria

#### 2.2.1. Introduction to the Sasang Constitution

According to the viewpoint of SCM, human beings have a tendency toward a skewed state by the seesaw balance between the visceral systems of a specific formulaic pair: the lung–liver pair and the spleen–kidney pair. Based on unbalanced states of these pairs of visceral systems, SCM classifies people into four constitutional types [14,15]. The TY type has a hyperactive lung system and a hypoactive liver system, whereas the TE type has a hyperactive liver system and a hypoactive lung system. On the other hand, the SY type has a hyperactive spleen system and a hypoactive kidney system, whereas the SE type has a hyperactive kidney system and hypoactive spleen system [14,15].

Lee Je-ma, who created Sasang theory, suggested the following factors to determine patients’ Sasang constitutions: physical appearance, features and way of speaking, temperament, physiological and pathological symptoms and pharmacology [24].

#### 2.2.2. Diagnosis of SC

SC was confirmed according to three conditions based on a previous study [22,25]. First, qualified experts with more than 5 years of clinical experience in the SCM field diagnosed SC types. Additionally, the participants were administered SC-specific pharmaceuticals before recruitment. If they experienced side effects on specific SC type-oriented herbal medicines or if the medicines did not have good effects, the subjects were not confirmed to have a specific SC type because herbal prescriptions that are unsuitable for different SC types could cause adverse effects. Only those who showed good effects with SC type-specific pharmaceuticals were confirmed to be a specific SC type.

#### 2.2.3. Pre-DM

Pre-DM was defined as a fasting plasma glucose (FPG) level between 100 and 125 mg/dL among individuals who were not undergoing treatment for diabetes in accordance with the ADA criteria [23].

#### 2.2.4. General Obesity

General obesity was defined as body mass index (BMI) ≥ 25 kg/m^2^, in accordance with the Asia-Pacific criteria of the World Health Organization guidelines [26].

### 2.3. Statistical Analysis

All participants were separated into normal and pre-DM groups to assess differences between the groups. T tests and chi-square tests were conducted to compare the general characteristics of the participants. The differences in FPG levels were compared using one-way ANOVA (Scheffé’s post hoc analysis) and *t* tests, and the prevalence of pre-DM was compared using the chi-square test according to SC type and obesity status, respectively. Multinomial logistic regression was performed to generate the odds ratios (ORs) and 95% confidence intervals (CIs) for the normal and pre-DM groups, and binary logistic regression was conducted for each SC type and among three SC types. The risk of pre-DM in the obese group was compared with that in the normal BMI group for each SC type, and the risk of pre-DM for the SY and TE types was compared with that of the SE type. Covariates, such as age, sex, blood pressure, triglycerides (TGs), and high-density lipoprotein cholesterol (HDL-C), were adjusted for to evaluate whether obesity could be a risk factor for pre-DM for each SC type and to see which SC type was a risk factor for pre-DM. The level of significance was set at a *p*-value  <  0.05, and all analyses were performed using SAS 9.4 software (SAS Institute Inc., Cary, NC, USA).

## 3. Results

### 3.1. Differences in General Characteristics

The numbers of male and female in this study were 483 and 1028 in the normal FPG group and 290 and 384 in the pre-DM group, respectively (*p* < 0.001). The numbers of individuals with SE, SY and TE types were 405, 516 and 590 in the normal FPG group and 133, 218 and 323 in the pre-DM group, respectively (*p* < 0.001). The numbers of people with normal BMI and those with obesity were 1049 and 462 in the normal FPG group and 396 and 278 in the pre-DM group, respectively (*p* < 0.001). The median FPG values were 87.8 ± 7.9 mg/dL in the normal FPG group and 108.0 ± 6.8 mg/dL in the pre-DM group (*p* < 0.001). The general characteristics, including age, BMI, blood pressure, TG levels and HDL-C levels, were significantly different between the normal and pre-DM groups. The details are described in Table 1.

### 3.2. FPG Values According to Sasang Constitution and Obesity Status

The median FPG values among individuals with the SE, SY and TE types were 87.6 ± 7.8, 87.2 ± 7.9 and 88.5 ± 7.8 in the normal FPG group and 107.2 ± 6.7, 108.3 ± 6.7, and 108.1 ± 6.9 in the pre-DM group, respectively. There was a significant difference in FPG values according to SC type in the normal FPG group (*p* < 0.05, SE < TE) and overall group (*p* < 0.001, SE, SY < TE).

The median FPG values among normal BMI and obese individuals were 87.4 ± 7.8 and 88.8 ± 8.0 in the normal FPG group and 108.0 ± 6.8 and 108.1 ± 6.9 in the pre-DM group, respectively. There was a significant difference in FPG values between obese and normal BMI individuals in the normal (*p* < 0.01) and overall groups (*p* < 0.001). The details are shown in Table 2.

### 3.3. Proportion of Normal and Prediabetic Individuals by SC Type in Stratified BMI Groups

In the normal BMI group, the proportion of individuals with pre-DM were 24%, 26.6%, and 32.8% for the SE, SY, and TE types, respectively. In the general obesity group, the proportion of individuals with pre-DM were 31.9%, 40.1%, and 37.3% for the SE, SY, and TE types, respectively. There was a significant difference in the proportion of each SC type among individuals with normal FPG levels and pre-DM in the normal BMI group (*p* = 0.013). However, there was no significant difference in the proportion of each SC type in the obese group. The details are shown in Figure 2.

The ORs of the TE type were significantly different from those of the SE type in the crude model (OR 1.667, 95% CI 1.314–2.115) and Model 1 (OR 1.353, 95% CI 1.031–1.776). However, the ORs of the TE type were not significantly different from those of the SE and SY types after adjusting for covariates in Model 2 in the overall group.

In addition, the ORs of the TE type were significantly different from those of the SE type in the crude model (OR 1.544, 95% CI 1.148–2.077) and Model 1 (OR 1.427, 95% CI 1.054–1.932). However, the ORs of the TE type were not significantly different from those of SE type after adjusting for covariates in Model 2 in the normal BMI group.

The ORs of the TE and SY types were not significantly different from those of the SE type regardless of covariates in the obese group. The details are shown in Table 3.

### 3.4. Proportion of Normal and Prediabetic Individuals Stratified by BMI in Each SC Type Group

The numbers of normal and pre-DM individuals were 373 (69.3%) and 118 (21.9%) in the normal BMI group and 32 (5.9%) and 15 (2.8%) in the obese group among individuals with the SE type, respectively. The numbers of normal and pre-DM individuals were 416 (56.7%) and 15 (20.6%) in the normal BMI group and 100 (13.6%) and 67 (9.1%) in the obese group among individuals with the SY type, respectively. The numbers of normal and pre-DM individuals were 260 (28.5%) and 127 (13.9%) in the normal BMI group and 330 (36.1%) and 196 (21.5%) in the obese group among individuals with the TE type, respectively.

There was a significant difference in the proportion of individuals with pre-DM and normal FPG levels between the obese and normal BMI groups among those with the SY type (*p* = 0.001). However, there was no significant difference in the proportion of individuals with pre-DM and normal FPG levels between the obese and normal BMI groups among individuals with the SE and TE types, respectively. The details are shown in Figure 3.

### 3.5. Adjusted ORs of the Obese Group for Pre-DM Compared with the Normal BMI Group among Individuals with SE, SY and TE Types

The ORs of the obese group were not significantly different from those of the normal BMI group in the crude model (OR 1.482, 95% CI 0.776–2.831), Model 1 (OR 1.303, 95% CI 0.673–2.523) or Model 2 (OR 1.096, 95% CI 0.553–2.17) among the SE types. For the SY type, the ORs of the obese group were significantly higher than those of the normal BMI group in the crude model (OR 1.846, 95% CI 1.286–2.649), Model 1 (OR 1.734, 95% CI 1.2–1.507) and Model 2 (OR 1.604, 95% CI 1.093–2.354). For the TE type, the ORs of the obese group were not significantly different from those of the normal BMI group in the crude model (OR 1.216, 95% CI 0.923–1.603), Model 1 (OR 1.201, 95% CI 0.907–1.589) or Model 2 (OR 1.042, 95% CI 0.777–1.397). The details are shown in Table 4.

## 4. Discussion

This study focused on specific SC types in combination with general obesity as risk factors for pre-DM, which is an important focus of preventive medicine.

We found that there was a significant difference in the proportion of SC types between individuals with pre-DM and individuals with normal FPG levels in the normal BMI group. In particular, the ORs of the TE type were significantly higher than those of SE in the crude model and Model 1 in the normal BMI group. This means that the TE type was a risk factor for pre-DM in the normal BMI group. Additionally, we found that there was a significant difference in the proportion of individuals with pre-DM and normal FPG levels between the obese and normal BMI groups only for the SY type. Furthermore, the ORs of the obese group were significantly higher than those of the normal BMI group only for the SY type. This means that if people with the SY type gain weight, they could have a significantly higher risk for pre-DM.

We collected clinical data from across the country and provided results that are representative of the Korean population with less local bias. The distribution of SC types was significantly different between the normal and pre-DM groups. The proportions of individuals with pre-DM and normal individuals were 35.4% and 64.6% in the TE group, 29.7% and 70.3% in the SY group and 24.7% and 75.3% in the SE group. The proportion of the TE type in the pre-DM group was relatively higher compared to those of other types. This implied that people with the TE type may be susceptible to pre-DM. Several studies have suggested that the TE type may act as a risk factor for metabolic syndrome [21], obesity [27], abdominal obesity [17], hypertension [18], etc.

The distributions of general obesity and normal BMI were significantly different between the normal FPG and pre-DM groups. General obesity assessed by BMI may be a risk factor for pre-DM, and some studies have demonstrated that obesity acts as a risk factor for pre-DM and DM [13]. Furthermore, the distributions of blood pressure, TGs, HDL-C, and FPG were significantly different between the normal and pre-DM groups, which revealed that the chronic metabolic index might also play a key role in pre-DM [28].

The FPG level for each SC type was significantly different between the normal FPG and pre-DM groups. However, the FPG levels among individuals with the TE type were significantly higher than those among individuals with the SE and SY types in the overall group. The FPG levels among individuals with the TE type were significantly higher than those among individuals with the SE type in the normal FPG group, but the FPG levels among individuals with the TE type were not significantly different from those among individuals with other SC types in the pre-DM group. Furthermore, there was a significant difference among SC types in the normal FPG group, but there was no difference among SC types in the pre-DM group. This finding suggested that the FPG levels among individuals with the TE type were higher than those among individuals with the SE type in the normal FPG group; however, there were no differences in the pre-DM group. This finding indicated that the FPG difference according to SC types in the overall group was attributed to the difference in FPG levels according to SC types in the normal FPG group but not in the pre-DM group. This difference may reveal physiological features of each SC type, not pathological features.

The FPG levels among individuals with general obesity and those with a normal BMI were significantly different between the normal and pre-DM groups. Furthermore, the FPG levels in the general obesity group were significantly higher than those in the normal BMI group overall. General obesity may act as a risk factor for T2D. In addition, the FPG levels among individuals with general obesity were significantly higher than those among normal BMI individuals in the normal FPG group, but there was no significant difference in the pre-DM group. This result suggested that the FPG levels of the normal BMI group could be physiological and that obesity itself may be a risk factor for pre-DM.

The FPG levels by SC type were significantly different between the normal and pre-DM groups. The FPG levels of individuals with the TE type were significantly higher than those of individuals with the SE and SY types overall. In addition, the FPG level of individuals with the TE type were significantly higher than those of individuals with the SE and SY types in the normal FPG group, but there was no significant difference according to SC type in the pre-DM group. This result suggested that the difference in FPG levels by SC type could be physiological and that SC type itself may be a potential factor for pre-DM. A prior blood pressure study also reported that the SC type could be physiological and that there were significant differences according to SC type in the normal blood pressure group but not in the hypertension group [29].

There was a significant difference in the proportion of SC types between the pre-DM and normal FPG groups, and the proportion of pre-DM among individuals with the TE type was high in the normal BMI group. However, there was no difference in the proportion according to SC type in the obese group. This means that SC, especially the TE type, could have played a key role in pre-DM in the normal BMI group but not in the obese group. This may suggest that the TE type may act as a risk factor for pre-DM and T2D. Several studies revealed that the TE type, independent of obesity, could be a risk factor for DM [30], and individuals with the TE type had a 1.4-fold higher prevalence of insulin resistance than individuals with the SY and SE types, suggesting that the TE type was a significant risk factor for insulin resistance [20].

We calculated ORs to determine whether specific SC types could be risk factors for pre-DM stratified by BMI. The ORs of the TE type were significantly higher than those of the SE type in the crude model and Model 1 in the overall group and especially the normal BMI group, but there was no difference in the obese group. Even though after adjusting for various variables, the difference disappeared, this implied that the TE type could be a potential risk factor for pre-DM, especially in the normal BMI group. Furthermore, the ORs of the TE and SY types were not significantly higher than those of the SE type in the overweight BMI group, which may suggest that obesity could play a more important role in pre-DM than the SC type. BMI was also associated with abnormal FPG levels. One study found that overweight/obesity was independently associated with impaired fasting glucose among adults and blood glucose between 100 and 125 mg/dL with no diabetic drug [31].

The distribution of FPG levels in the general obesity and normal BMI groups was not significantly different among individuals with the TE and SE types, respectively. However, the distribution was significantly different among individuals with the SY type. This result suggests that the SY type combined with general obesity could be a risk factor for pre-DM, whereas the other types were not risk factors.

We calculated ORs to determine whether specific SC types combined with general obesity could be risk factors for pre-DM. In the results, the ORs of the obese group were significantly higher than those of the normal BMI group in the crude and adjusted models only among individuals with the SY type. This finding revealed that the SY type combined with general obesity is associated with increased susceptibility to pre-DM and that other SC types combined with general obesity are not associated with increased susceptibility to pre-DM. This result suggests that SC types could play an important role in traditional preventive care. SCM states that a patient’s susceptibility to pathologies differs by SC. Accordingly, this also applies to DM [19]. DM and pre-DM were possibly associated with So-gal among individuals with the SY type and Jo-yul among individuals with the TE type based on the hypothetical interpretation of SCM theory. Lee insisted that these constitutional diseases came from the imbalance of organs and intestines of the body. In addition, the lungs of the TE type are hypoactive, whereas the liver of the TE type is hyperactive. Therefore, the TE type is characterized by a state of weak consumption and strong storage of Qi and body fluid [14,15]. This is a potential mechanism by which the TE type in the normal BMI group could be a risk factor for pre-DM. On the other hand, the SY type has a hyperactive spleen system and a hypoactive kidney system, which leads to a consistent state of strong raw material intake and weak waste discharge. This is a potential mechanism by which the SY type with general obesity could be a risk factor for pre-DM [14,15,20].

This study had several strengths. First, local bias may have less of an effect on the results because the subjects were recruited from nationwide South Korean medical centers, and the data collection was performed by following SOPs [22]. Second, this study aimed to improve preventive care by focusing on pre-DM stages, which are becoming popular in health promotion.

This study also had a few limitations. As the design of this study was cross-sectional, we could not confirm a causal relationship between each SC type combined with obesity and pre-DM; therefore, the evidence level of this result could be relatively low. Although we considered the influences of potential variables such as age, sex, and blood pressure, we did not control for all of the risk factors that influence pre-DM. As FPG was measured only once and habitual life and environmental factors possibly affect FPG, the results might be less rigorous. Epigenetic mechanisms impact gene expression that could predispose individuals to the diabetic phenotype during intrauterine and early postnatal development, as well as throughout adult life [32]. Accordingly, further studies with longer-term follow-up designs with epigenetic conditions are needed to confirm SC type combined with obesity as an important risk factor for pre-DM.

## 5. Conclusions

There was a significant difference in the proportion of each SC type between individuals with pre-DM and those with normal FPG levels in the normal BMI group. There was also a significantly different distribution between the normal and pre-DM groups in terms of the SY type. Furthermore, the ORs of the TE type were significantly different from those of the SE type, especially in the normal BMI group, and the ORs of the obese group were significantly higher than those of the normal BMI group among individuals with the SY type. This finding reveals that SC combined with obesity is significantly associated with pre-DM and might be a significant risk factor for pre-DM. SC should be considered when treating the disease. Further study including epigenetic is needed.

## Figures and Tables

**Figure 1 healthcare-10-02286-f001:**
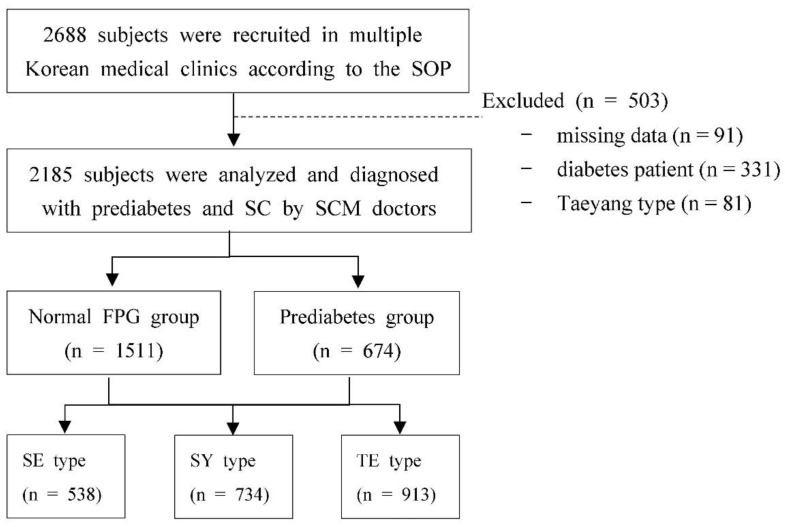
Flowchart of the study. SOP: Standard operating procedure, SC: Sasang constitution, SCM: Sasang constitutional medicine, FPG: fasting plasma glucose, SE: Soeumin, SY: Soyangin, TE: Taeeumin.

**Figure 2 healthcare-10-02286-f002:**
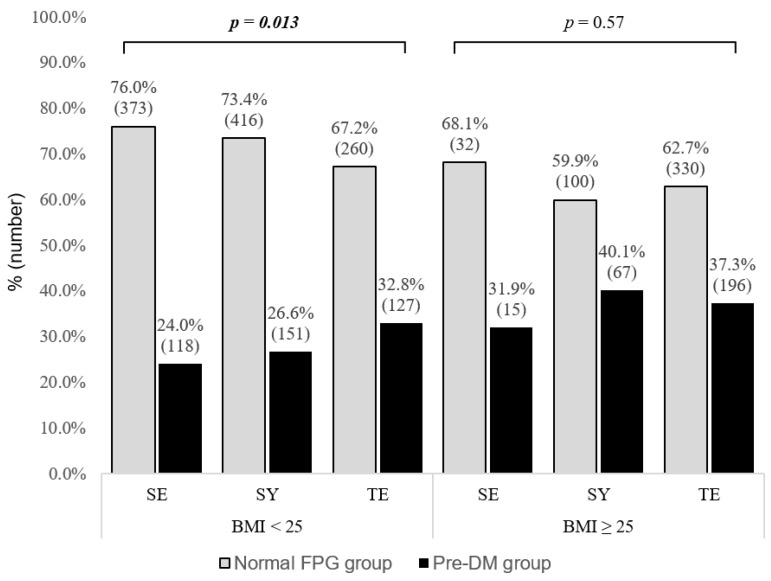
Proportion of individuals with pre-DM by SC type according to BMI. BMI: body mass index, SE: Soeumin, SY: Soyangin, TE: Taeeumin, FPG: Fasting plasma glucose. Boldface type indicates statistical significance.

**Figure 3 healthcare-10-02286-f003:**
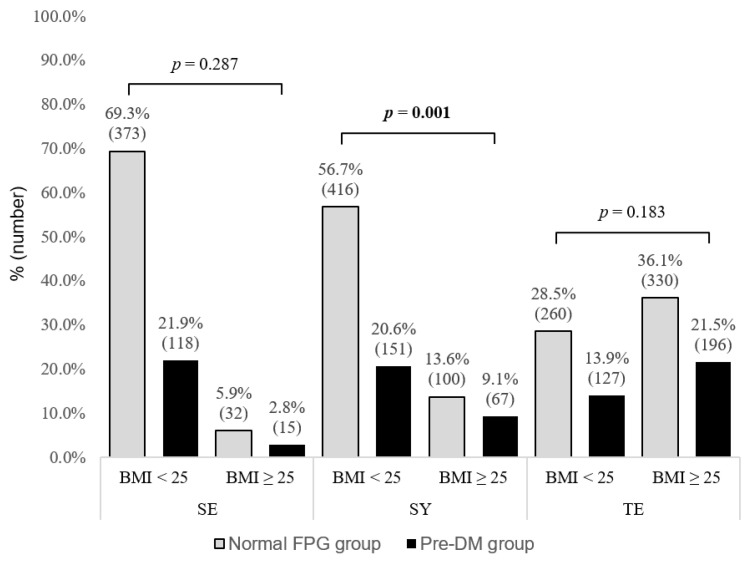
Proportion of normal individuals and individuals with prediabetes stratified by BMI among individuals with SE, SY and TE types. BMI: body mass index, SE: Soeumin, SY: Soyangin, TE: Taeeumin, FPG: Fasting plasma glucose. Boldface type indicates statistical significance.

**Table 1 healthcare-10-02286-t001:** General characteristics.

	Normal FPG Group (n = 1511)	Pre-DM Group (n = 674)	Total (n = 2185)	*p* Value
Sex (N, %)				
Male	483 (62.5)	290 (37.5)	773 (100)	<0.001
Female	1028 (72.8)	384 (27.2)	1412 (100)	
Age (years)	54.9 ± 9.9	57.9 ± 10.2	55.8 ± 10.1	<0.001
SC type (N, %)				
SE	405 (75.3)	133 (24.7)	538 (100)	<0.001
SY	516 (70.3)	218 (29.7)	734 (100)	
TE	590 (64.6)	323 (35.4)	913 (100)	
BMI (kg/m^2^)	23.7 ± 3.1	24.3 ± 3.1	23.9 ± 3.1	<0.001
BMI < 25	1049 (72.6)	396 (27.4)	1445 (100)	<0.001
BMI ≥ 25	462 (62.4)	278 (37.6)	740 (100)	
FPG (mg/dL)	87.8 ± 7.9	108.0 ± 6.8	94 ± 12.0	<0.001
Systolic BP (mmHg)	120.3 ± 15.2	124.8 ± 15.8	121.7 ± 15.5	<0.001
Diastolic BP (mmHg)	77.6 ± 10.8	80 ± 11.6	78.4 ± 11.1	<0.001
TGs (mg/dL)	128.6 ± 77.8	147.9 ± 95.2	134.5 ± 84	<0.001
HDL-C (mg/dL)	47.5 ± 12.4	44.8 ± 11.7	46.6 ± 12.3	<0.001

*p* value: Fisher’s exact test or chi-square test for categorical variables, *t* test for continuous variables between the normal and pre-DM groups. SC: Sasang constitution, SE: Soeumin, SY: Soyangin, TE: Taeeumin, BMI: body mass index, FPG: fasting plasma glucose, BP: blood pressure, TGs: triglycerides, HDL-C: high-density lipoprotein cholesterol; Pre-DM: prediabetes mellitus.

**Table 2 healthcare-10-02286-t002:** FPG values by Sasang constitution type and obesity status.

	Normal FPG Group	Pre-DM Group	Total	*p* Value ^a^
SC type				
SE	87.6 ± 7.8	107.2 ± 6.7	92.4 ± 11.3	<0.001
SY	87.2 ± 7.9	108.3 ± 6.7	93.4 ± 12.3	<0.001
TE	88.5 ± 7.8	108.1 ± 6.9	95.4 ± 12.0	<0.001
*p* value ^b^	0.014 ^(SE < TE)^	0.36	<0.001 ^(SE, SY < TE)^	
BMI				
BMI < 25	87.4 ± 7.8	108.0 ± 6.8	93.0 ± 11.9	<0.001
BMI ≥ 25	88.8 ± 8.0	108.0 ± 6.8	96.0 ± 12.0	<0.001
*p* value ^b^	0.002	0.96	<0.001	

^a^ Comparison of FPG between the normal and pre-DM groups by SC type and obesity status (row). ^b^ Comparison of FPG values among people of different SC types and obesity status within the normal and pre-DM groups (column). SC: Sasang constitution, BMI: body mass index, SE: Soeumin, SY: Soyangin, TE: Taeeumin, Pre-DM: prediabetes mellitus.

**Table 3 healthcare-10-02286-t003:** Adjusted ORs (95% CI) of the SY and TE types compared with those of the SE type for pre-DM considering BMI.

	SE	SY		TE	
		ORs (95% CIs)	*p* Value	ORs (95% CIs)	*p* Value
BMI < 25 ^a^					
Crude	Reference	1.147 (0.869–1.516)	0.333	1.544 (1.148–2.077)	0.004
Model 1	Reference	1.122 (0.845–1.489)	0.427	1.427 (1.054–1.932)	0.021
Model 2	Reference	1.095 (0.822–1.457)	0.536	1.354 (0.995–1.844)	0.054
BMI ≥ 25 ^a^					
Crude	Reference	1.429 (0.719–2.841)	0.308	1.267 (0.669–2.399)	0.467
Model 1	Reference	1.355 (0.676–2.716)	0.392	1.205 (0.632–2.299)	0.571
Model 2	Reference	1.355 (0.671–2.734)	0.397	1.158 (0.604–2.221)	0.659
Total ^b^					
Crude	Reference	1.286 (1.0–1.655)	0.05	1.667 (1.314–2.115)	<0.001
Model 1	Reference	1.187 (0.916–1.54)	0.195	1.353 (1.031–1.776)	0.029
Model 2	Reference	1.161 (0.894–1.509)	0.263	1.281 (0.973–1.686)	0.077

^a^ Model 1 was adjusted for age and sex, and Model 2 was adjusted for age, sex, systolic and diastolic blood pressure, TGs, and HDL-C by stratified BMI. ^b^ Model 1 was adjusted for age, sex, and BMI, and Model 2 was adjusted for age, sex, BMI, systolic and diastolic blood pressure, TGs, and HDL-C in the overall group. Pre-DM: prediabetes mellitus, SE: Soeumin, SY: Soyangin, TE: Taeeumin, BMI: Body mass index.

**Table 4 healthcare-10-02286-t004:** Adjusted ORs (95% CIs) of the obese group compared with the normal BMI group for pre-DM among individuals with SE, SY and TE types.

	ORs (95% CIs) for Pre-DM	*p* Value
SE		
BMI < 25	Reference	
Crude	1.482 (0.776–2.831)	0.234
Model 1	1.303 (0.673–2.523)	0.432
Model 2	1.096 (0.553–2.17)	0.793
SY		
BMI < 25	Reference	
Crude	1.846 (1.286–2.649)	0.001
Model 1	1.734 (1.2–2.507)	0.003
Model 2	1.604 (1.093–2.354)	0.016
TE		
BMI < 25	Reference	
Crude	1.216 (0.923–1.603)	0.165
Model 1	1.201 (0.907–1.589)	0.201
Model 2	1.042 (0.777–1.397)	0.785

Model 1 was adjusted for age and sex. Model 2 was adjusted for age, sex, systolic and diastolic blood pressure, TGs, and HDL-C. Pre-DM: prediabetes mellitus, SE: Soeumin, SY: Soyangin, TE: Taeeumin, BMI: Body mass index.

## Data Availability

The data presented in this study are available upon request from the corresponding author. The data are not publicly available due to privacy or ethical restrictions.

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
