# Peer review of "Sasang Constitution Type Combined with General Obesity May Act as a Risk Factor for Prediabetes Mellitus"

_healthcare, 2022, doi:10.3390/healthcare10112286_

Round 1

Reviewer 1 Report

This interesting study shows the use of fenotype in the prediction of prediabetes. I have the following remarks:

-          The importance of the patient's fenotype suggests that epigenetics are involved, which should be noted in the Discussion.

-          I notice a disproportional number of women in the study, despite diabetes being a typical "male" disease. Where any differences between men and women?

Author Response

Response to Reviewer 1 Comments

We sincerely thank the reviewer for their effort and time in reviewing our manuscript. Detailed point-by-point responses and a list of changes to the revised manuscript are provided below. Your comments have improved the overall quality of the manuscript. All of the authors truly appreciate your efforts and assistance.

This interesting study shows the use of fenotype in the prediction of prediabetes. I have the following remarks:

Point 1: The importance of the patient's fenotype suggests that epigenetics are involved, which should be noted in the Discussion.

Response 1: Several studies have shown that epigenetic conditions, such as lifestyle, diet, and exercise, affect the risk factors for diabetes mellitus and pre-DM. This study adjusted for several diseases but did not consider acquired health factors. Therefore, we described this limitation in the Discussion section (page 11, lines 448-452).

  • Epigenetic mechanisms impact gene expression that could predispose individuals to the diabetic phenotype during intrauterine and early postnatal development, as well as throughout adult life [32]. Accordingly, further studies with longer-term follow-up designs with epigenetic conditions are needed to confirm SC type combined with obesity as an important risk factor for pre-DM.

Point 2: I notice a disproportional number of women in the study, despite diabetes being a typical "male" disease. Where any differences between men and women?

Response 2: As you mentioned, diabetes is a typical "male" disease. This study also revealed that the proportion of diabetes among males (37.5%) was higher than that among females (27.2%) (p < 0.001). Most likely, as we expressed vertical %, this might have misled you. To clarify this, we used horizontal % (page 4, revised Table 1).

Reviewer 2 Report

The manuscript submitted by Baek et al aims to assess the risk of pre-diabetes (pre-DM) according to Sasang constitution and obesity status. The work is original and is of interest, the methodology and introduction are appropriate. However there some major issues that should be addressed.

Abstract:

1.     Please provide few words of explanation in the abstract on the Sasang constitution.

2.     Please state in the abstract the type of study and the time interval when data was collected/participants were enrolled.

Materials and methods

3.     Please clearly state the inclusion and the exclusion criteria for the study participation. Why were enrolled only young adults?

4.     How were participants recruited? Consecutive participants or other recruitment method?

5.     Please provide a brief description on how Sasang constitutional types are assessed in the traditional Korean medicine and the characteristics of each type. These will help the readers that are not familiar with traditional Korean medicine.

6.     Reference 26 does not seem suitable as it does not provide a description on how Sasang constitution is assessed.

Results

7.     Please replace "normal group" with "normal fasting plasma glucose group" here and thought the manuscript.

8.     In Figure 2, more important is the % of participants with pre-DM in each constitutional type than the number of participants. Please adjust the figure as now is misleading.

9.     Please compare the % of participants with preDM by BMI in each constitutional type  (for example 21.9% vs. 20.6% vs. 13.9%, p=?). According to figure 2 it looks like the % varies according to constitutional type in the same BMI category.

10.  Please re-write section 3.4. The authors did not compare ORs between participants with and without obesity. They assessed whether obesity was associated with higher risk of preDM in various Sasang types and this can be seen in the OR value.

11.  The proportion of preDM is higher in those with normal BMI in the SE and SY types as compared to TE group. Also, it increases in those with overweight/obesity from SE to SY and to TE subtypes. It would be interesting to see whether these types are risk factors for preDM irrespective of BMI. Please check whether SC types per se are associated with an increased risk of the presence of pre-DM. For this the authors should perform another logistic regression with SC types as predictors and adjust for obesity status.

Discussion section

12.  Please start the Discussions with the main findings of this manuscript and comment them in the context of previous literature. Now is difficult to find these in the Discussion section.

13.  The % of participants with TE type is significantly higher in the preDM group than in the normal FPG group. Please adjust the third paragraph of the discussion section to reflect this and add references to support the statement on TE acting as risk factor for metabolic syndrome.

14.  The authors state that “TE type may act as a risk factor for T2D”. This statement cannot be supported by the current results. It can be added (or a similar statement) only if the additional regression analysis would confirm that Sasang types are risk factors for DM (please see comment above).

15.  Paragraph lines 270-274 is redundant. Please remove it.

16.  Paragraph lines 275-278 should be re-written as currently is difficult to read. No difference in the FPG was observed according to SC types and BMI groups in participants with preDM. However, the authors observed higher FPG in those with TE type and high BMI in the normal FPG group. Please comment on this. Is this and expected finding? Any previous research in the literature showed that TE type was associated with higher FPG or risk factors for it? If yes please briefly describe the previous findings (population, type of research, findings). Similar should be done for BMI categories.

17.  Paragraph lines 284-291. Is there any potential mechanism linking SY type, obesity and preDM/diabetes? Please comment in the context of previous literature on this subject.

Author Response

Response to Reviewer 2 Comments

We sincerely thank the reviewer for their effort and time in reviewing our manuscript. Detailed point-by-point responses and a list of changes to the revised manuscript are provided below. Your comments have improved the overall quality of the manuscript. All of the authors truly appreciate your efforts and assistance.

The manuscript submitted by Baek et al aims to assess the risk of pre-diabetes (pre-DM) according to Sasang constitution and obesity status. The work is original and is of interest, the methodology and introduction are appropriate. However there some major issues that should be addressed.

[Abstract]

Point 1: Please provide few words of explanation in the abstract on the Sasang constitution.

Response 1: Per your comment, we added an explanation for the Sasang constitution (page 1, lines 9-10).

  • Sasang constitutional medicine is a traditional customized medicine in Korea that classifies people into four types: Taeeumin (TE), Taeyangin (TY), Soeumin (SE), and Soyangin (SY).

Point 2: Please state in the abstract the type of study and the time interval when data was collected/participants were enrolled.

Response 2: Per your comment, we added the type of study and the time interval (page 1, lines 12-13).

  • This study was cross-sectional and was conducted from Nov. 2007 to Jul. 2011 in 23 Korean medical clinics.

[Materials and methods]

Point 3: Please clearly state the inclusion and the exclusion criteria for the study participation. Why were enrolled only young adults?

Point 4: How were participants recruited? Consecutive participants or other recruitment method?

Response 3 and 4: Per your comment, we state the inclusion and exclusion criteria and recruitment method for the study participation in 2.1. Data source and subjects. As there was an error in ages (the average age of the participants was in the 50s), we deleted and corrected this (page 2, lines 80-85).

  • The subjects included those who voluntarily agreed to participate in this study among adults who met the criteria for taking SC-specific pharmaceuticals. The exclusion criteria included subjects who were physically unable to follow the instructions of the researcher, those who had a deformation in the measurement location, and pregnant women. Subjects were consecutively recruited via posts on both online and offline boards during the study period [22].

Point 5: Please provide a brief description on how Sasang constitutional types are assessed in the traditional Korean medicine and the characteristics of each type. These will help the readers that are not familiar with traditional Korean medicine.

Response 5: Per your comment, we provide a brief description of how Sasang constitutional types are assessed in traditional Korean medicine and the characteristics of each type (page 3, lines 123-134).

  • 2.1. Introduction to the Sasang Constitution

According to the viewpoint of SCM, human beings have a tendency toward a skewed state by the seesaw balance between the visceral systems of a specific formulaic pair: the lung–liver pair and the spleen–kidney pair. Based on unbalanced states of these pairs of visceral systems, SCM classifies people into four constitutional types [14,15]. The TY type has a hyperactive lung system and a hypoactive liver system, whereas the TE type has a hyperactive liver system and a hypoactive lung system. On the other hand, the SY type has a hyperactive spleen system and a hypoactive kidney system, whereas the SE type has a hyperactive kidney system and hypoactive spleen system [14,15].

Lee Je-ma, who created Sasang theory, suggested the following factors to determine patients’ Sasang constitutions: physical appearance, features and way of speaking, temperament, physiological and pathological symptoms and pharmacology [24].

Point 6: Reference 26 does not seem suitable as it does not provide a description on how Sasang constitution is assessed.

Response 6: Per your comment, we added a suitable reference, no. 22 (page 3, line 136).

  • Jin, H. J.; Baek, Y.; Kim, H. S.; Ryu, J.; Lee, S. Constitutional multicenter bank linked to Sasang constitutional phenotypic data. BMC Complement Altern Med. 2015, 15(1), 46, doi.org/10.1186/s12906-015-0553-3.

[Results]

Point 7: Please replace "normal group" with "normal fasting plasma glucose group" here and thought the manuscript.

Response 7: To clarify the meaning, we changed ‘normal group’ to ‘normal fasting plasma glucose group’, and ‘nonobese group’ to ‘normal BMI group’ throughout the manuscript.

Point 8: In Figure 2, more important is the % of participants with pre-DM in each constitutional type than the number of participants. Please adjust the figure as now is misleading.

Response 8: Per your comment, we added the % of participants in Fig. 3 (page 8, line 295).

Point 9: Please compare the % of participants with preDM by BMI in each constitutional type (for example 21.9% vs. 20.6% vs. 13.9%, p=?). According to figure 2 it looks like the % varies according to constitutional type in the same BMI category.

Response 9: Per your comment, we performed further analysis as the % of participants with pre-DM by BMI for each constitutional type and described the content (page 6, lines 233-240, added Fig. 2).

  • 3. Proportion of normal and prediabetic individuals by SC type in stratified BMI groups

In the normal BMI group, the proportion of individuals with pre-DM were 24%, 26.6%, and 32.8% for the SE, SY, and TE types, respectively. In the general obesity group, the proportion of individuals with pre-DM were 31.9%, 40.1%, and 37.3% for the SE, SY, and TE types, respectively. There was a significant difference in the proportion of each SC type among individuals with normal FPG levels and pre-DM in the normal BMI group (p = 0.013). However, there was no significant difference in the proportion of each SC type in the obese group. The details are shown in Fig. 2.

Fig. 2. Proportion of individuals with pre-DM by SC type according to BMI

Point 10: Please re-write section 3.4. The authors did not compare ORs between participants with and without obesity. They assessed whether obesity was associated with higher risk of preDM in various Sasang types and this can be seen in the OR value.

Point 11: The proportion of preDM is higher in those with normal BMI in the SE and SY types as compared to TE group. Also, it increases in those with overweight/obesity from SE to SY and to TE subtypes. It would be interesting to see whether these types are risk factors for preDM irrespective of BMI. Please check whether SC types per se are associated with an increased risk of the presence of pre-DM. For this the authors should perform another logistic regression with SC types as predictors and adjust for obesity status.

Response 10 and 11: We think your comment is valid and appropriate. Per your comment, we performed further analysis of the proportion and adjusted ORs of individuals with pre-DM by SC type and BMI. To determine whether obesity was associated with a higher risk of pre-DM for various Sasang types and to check whether SC types were associated with an increased risk of the presence of pre-DM, we added another analysis (page 6, lines 247-257, Table 3 added).

  • The ORs of the TE type were significantly different from those of the SE type in the crude model (OR 1.667, 95% CI 1.314-2.115) and Model 1 (OR 1.353, 95% CI 1.031-1.776). However, the ORs of the TE type were not significantly different from those of the SE and SY types after adjusting for covariates in Model 2 in the overall group.

In addition, the ORs of the TE type were significantly different from those of the SE type in the crude model (OR 1.544, 95% CI 1.148-2.077) and Model 1 (OR 1.427, 95% CI 1.054-1.932). However, the ORs of the TE type were not significantly different from those of SE type after adjusting for covariates in Model 2 in the normal BMI group.

The ORs of the TE and SY types were not significantly different from those of the SE type regardless of covariates in the obese group.

Table 3. Adjusted ORs (95% CI) of the SY and TE types compared with those of the SE type for pre-DM considering BMI.

SE

SY

TE

ORs (95% CIs)

p value

ORs (95% CIs)

p value

BMI < 25a

Crude

Reference

1.147 (0.869-1.516)

0.333

1.544 (1.148-2.077)

0.004

Model 1

Reference

1.122 (0.845-1.489)

0.427

1.427 (1.054-1.932)

0.021

Model 2

Reference

1.095 (0.822-1.457)

0.536

1.354 (0.995-1.844)

0.054

BMI ≥ 25a

Crude

Reference

1.429 (0.719-2.841)

0.308

1.267 (0.669-2.399)

0.467

Model 1

Reference

1.355 (0.676-2.716)

0.392

1.205 (0.632-2.299)

0.571

Model 2

Reference

1.355 (0.671-2.734)

0.397

1.158 (0.604-2.221)

0.659

Total b

Crude

Reference

1.286 (1.0-1.655)

0.05

1.667 (1.314-2.115)

<0.001

Model 1

Reference

1.187 (0.916-1.54)

0.195

1.353 (1.031-1.776)

0.029

Model 2

Reference

1.161 (0.894-1.509)

0.263

1.281 (0.973-1.686)

0.077

[Discussion section]

Point 12: Please start the Discussions with the main findings of this manuscript and comment them in the context of previous literature. Now is difficult to find these in the Discussion section.

Response 12: Per your comment, we revised the main findings of this manuscript in the first part of the Discussion (page 9, lines 330-339).

  • We found that there was a significant difference in the proportion of SC types be-tween individuals with pre-DM and individuals with normal FPG levels in the normal BMI group. In particular, the ORs of the TE type were significantly higher than those of SE in the crude model and Model 1 in the normal BMI group. This means that the TE type was a risk factor for pre-DM in the normal BMI group. Additionally, we found that there was a significant difference in the proportion of individuals with pre-DM and normal FPG levels between the obese and normal BMI groups only for the SY type. Furthermore, the ORs of the obese group were significantly higher than those of the normal BMI group only for the SY type. This means that if people with the SY type gain weight, they could have a significantly higher risk for pre-DM.

Point 13: The % of participants with TE type is significantly higher in the preDM group than in the normal FPG group. Please adjust the third paragraph of the discussion section to reflect this and add references to support the statement on TE acting as risk factor for metabolic syndrome.

Response 13: Per your comment, we adjusted this and added a reference (page 9, lines 344-348).

  • The proportion of the TE type in the pre-DM group was relatively higher compared to those of other types. This implied that people with the TE type may be susceptible to pre-DM. Several studies have suggested that the TE type may act as a risk factor for meta-bolic syndrome [21], obesity [27], abdominal obesity [17], hypertension [18], etc.

Point 14: The authors state that “TE type may act as a risk factor for T2D”. This statement cannot be supported by the current results. It can be added (or a similar statement) only if the additional regression analysis would confirm that Sasang types are risk factors for DM (please see comment above).

Response 14: Per your comment, we added an additional regression analysis (Table 3) and added a discussion (page 10, lines 400-411).

  • We calculated ORs to determine whether specific SC types could be risk factors for pre-DM stratified by BMI. The ORs of the TE type were significantly higher than those of the SE type in the crude model and Model 1 in the overall group and especially the normal BMI group, but there was no difference in the obese group. Even though after adjusting for various variables, the difference disappeared, this implied that the TE type could be a po-tential risk factor for pre-DM, especially in the normal BMI group. Furthermore, the ORs of the TE and SY types were not significantly higher than those of the SE type in the over-weight BMI group, which may suggest that obesity could play a more important role in pre-DM than the SC type. BMI was also associated with abnormal FPG levels. One study found that overweight/obesity was independently associated with impaired fasting glu-cose among adults and blood glucose between 100 and 125 mg/dl with no diabetic drug [31].

Point 15: Paragraph lines 270-274 is redundant. Please remove it.

Response 15: Per your comment, we removed these paragraph lines.

Point 16: Paragraph lines 275-278 should be re-written as currently is difficult to read. No difference in the FPG was observed according to SC types and BMI groups in participants with preDM. However, the authors observed higher FPG in those with TE type and high BMI in the normal FPG group. Please comment on this. Is this and expected finding? Any previous research in the literature showed that TE type was associated with higher FPG or risk factors for it? If yes please briefly describe the previous findings (population, type of research, findings). Similar should be done for BMI categories.

Response 16: As you said, no difference in FPG was observed according to SC type or BMI group among participants with pre-DM, but there were higher FPG levels among those with the TE type and high BMI in the overall and normal FPG groups (Table 2). We added another analysis and revised the discussion of this part (page 10, lines 380-399).

  • The FPG levels by SC type were significantly different between the normal and pre-DM groups. The FPG levels of individuals with the TE type were significantly higher than those of individuals with the SE and SY types overall. In addition, the FPG level of individuals with the TE type were significantly higher than those of individuals with the SE and SY types in the normal FPG group, but there was no significant difference accord-ing to SC type in the pre-DM group. This result suggested that the difference in FPG levels by SC type could be physiological and that SC type itself may be a potential factor for pre-DM. A prior blood pressure study also reported that the SC type could be physiologi-cal and that there were significant differences according to SC type in the normal blood pressure group but not in the hypertension group [29].

There was a significant difference in the proportion of SC types between the pre-DM and normal FPG groups, and the proportion of pre-DM among individuals with the TE type was high in the normal BMI group. However, there was no difference in the propor-tion according to SC type in the obese group. This means that SC, especially the TE type, could have played a key role in pre-DM in the normal BMI group but not in the obese group. This may suggest that the TE type may act as a risk factor for pre-DM and T2D. Several studies revealed that the TE type, independent of obesity, could be a risk factor for DM [30], and individuals with the TE type had a 1.4-fold higher prevalence of insulin re-sistance than individuals with the SY and SE types, suggesting that the TE type was a sig-nificant risk factor for insulin resistance [20].

Point 17: Paragraph lines 284-291. Is there any potential mechanism linking SY type, obesity and preDM/diabetes? Please comment in the context of previous literature on this subject.

Response 17: Per your comment, we added a potential mechanism linking the TE and SY types (page 11, lines 424-436).

  • SCM states that a patient’s susceptibility to pathologies differs by SC. Accordingly, this also applies to DM [19]. DM and pre-DM were possibly associated with So-gal among in-dividuals with the SY type and Jo-yul among individuals with the TE type based on the hypothetical interpretation of SCM theory. Lee insisted that these constitutional diseases came from the imbalance of organs and intestines of the body. In addition, the lungs of the TE type are hypoactive, whereas the liver of the TE type is hyperactive. Therefore, the TE type is characterized by a state of weak consumption and strong storage of Qi and body fluid [14,15]. This is a potential mechanism by which the TE type in the normal BMI group could be a risk factor for pre-DM. On the other hand, the SY type has a hyperactive spleen system and a hypoactive kidney system, which leads to a consistent state of strong raw material intake and weak waste discharge. This is a potential mechanism by which the SY type with general obesity could be a risk factor for pre-DM [14,15,20].

Round 2

Reviewer 2 Report

Thank you for adressing all comments. I do not have any further comment on the manuscript.